# TEM8/ANTXR1-specific CAR T cells mediate toxicity in vivo

**Kristina Petrovic**[1¤a], **Joseph Robinson**[1¤b], **Katharine Whitworth**[1], **Elizabeth Jinks**[1], **Abeer Shaaban**[2], **Steven P. Lee**[1]*

**1** Institute of Immunology and Immunotherapy, University of Birmingham, Birmingham, United Kingdom,
**2** Histopathology Department, Queen Elizabeth Hospital, Birmingham, United Kingdom

¤a Current address: Immunocore, Abingdon, Oxfordshire, United Kingdom.
¤b Current address: Achilles Therapeutics, Stevenage, Hertfordshire, United Kingdom.
* s.p.lee@bham.ac.uk

**Data Availability Statement:** All relevant data are within the manuscript and its Supporting Information files.

**Funding:** This work was funded by a Wellcome PhD studentship award to KP (grant 102286/Z/13/

## Abstract

Engineering T-cells to express receptors specific for antigens present on tumour tissue is proving a highly effective treatment for some leukaemias. However, extending this to solid tumours requires antigens that can be safely and effectively targeted. TEM8, a marker over-expressed on the vasculature of some solid tumours, has been proposed as one such target. A recent report stated that T-cells engineered to express a TEM8-specific chimeric antigen receptor (CAR), when injected into mouse models of triple negative breast cancer, are both safe and effective in controlling tumour growth. Here we report contrasting data with a panel of TEM8-specific CAR-T-cells including one generated from the same antibody used in the other study. We found that the CAR-T-cells demonstrated clear TEM8-specific cytotoxic and cytokine release responses in vitro, but when injected into healthy C57BL6 and NSG mice they rapidly and selectively disappeared from the circulation and in most cases caused rapid toxicity. Infusing CAR-T-cells into a TEM8-knockout mouse indicated that selective loss of cells from the circulation was due to targeting of TEM8 in healthy tissues. Histological analysis of mice treated with a TEM8-specific CAR revealed evidence of inflammation in the lung and spleen with large collections of infiltrating neutrophils. Therefore our data raise concerns over potential on-target off-tumour toxicity with CARs targeting TEM8 and these should be considered carefully before embarking upon clinical trials with such agents.

## Introduction

Adoptive therapy using tumour-specific T-cells can be a very effective treatment for human cancer, but naturally occurring T-cells with the appropriate tumour specificity are rare. Therefore more recent work has employed genetic engineering techniques to rapidly and reliably introduce genes encoding receptors specific for defined tumour antigens[1]. This includes engineering T-cells to induce expression of a chimeric antigen receptor (CAR), which generally combines the antigen-binding domains of an antibody in the form of a single chain variable fragment (scFv) linked to the signalling domain (CD3ζ chain) from the T-cell receptor complex. Such CARs based on an antibody specific for the B cell marker CD19 have proven

Z). https://wellcome.ac.uk/. The funders had no role in study design, data collection and analysis, decision to publish, or preparation of the manuscript.

**Competing interests:** S. Lee is listed as an inventor on the following patent application/patent families; PCT/GB2017/050686 (published as WO2017/158337) and PCT/GB2017/050689 (WO2017/158339), and has interests in anti-CLEC14A CAR-T cells, studies relating to which have been supported by Cell and Gene Therapy Catapult UK and licensed to Chimeric Therapeutics Ltd. Note this does not alter our adherence to PLOS ONE policies on sharing data and materials.

highly effective in treating some leukaemias[2–4], leading to recent FDA approval for some of these therapies. Unsurprisingly, these CD19-specific CARs mediate so called "on-target, off-tumour" effects since the target antigen is also expressed on healthy B cells leading to B-cell aplasia and hypogammaglobulinaemia, but this can be managed clinically by regular infusions with immunoglobulin.

Given the clinical success of CAR T-cell therapy for leukaemias, there is considerable interest in extending its use to the more common solid tumours. However, this is proving more challenging, partly because of the hostile tumour microenvironment that can include multiple immune evasion mechanisms but also because of the lack of suitable target antigens. Targeting the tumour stroma such as the tumour vasculature, rather than the malignant cells directly, is an attractive alternative approach since it is readily accessed by circulating T-cells and is less genetically unstable than malignant cells[5], reducing the likelihood of antigen-loss variants [6]. Furthermore, targeting the tumour vasculature should not only damage the surrounding tumour tissue but also malignant tissue downstream of that vessel. This approach again requires the identification of specific antigens, and a growing list of tumour endothelial markers (TEMs) have been described.

TEM8 was originally identified as a TEM in colorectal carcinoma[7] and although it is expressed in the endothelial cells of developing mouse embryos[8], it was not detectable in healthy tissues of adult mice[9]. Equally, studies in human tissues failed to detect it during physiological angiogenesis required for corpus luteum development and wound healing[7]. TEM8 is a single-pass cell surface glycoprotein 564 amino acids long, with a von Willebrand factor type A (vWA) domain in its extracellular region[10]. It is highly conserved with 96% amino acid sequence identity between mouse and man[8]. TEM8 binds collagen and promotes endothelial cell migration in vitro and is thereby thought to play an important role in angiogenesis[11, 12]. TEM8 also acts as a receptor for anthrax toxin[13], and is known as anthrax toxin receptor 1 (ANTXR1).

In addition to colorectal cancer, TEM8 is upregulated on vessels in various human and mouse tumour types[8, 11, 14], and in some cases is also expressed by the malignant cells[8, 15, 16]. Its importance in cancer has been demonstrated using TEM8 knockout mice that show impaired tumour growth[9, 17]. Furthermore, targeting this molecule in mice using TEM8 vaccines[18, 19], an anti-TEM8/truncated tissue factor fusion protein[14] and sublethal doses of anthrax toxin[20] can inhibit angiogenesis and tumour growth, as well as prolong survival. In 2012, a study using antibody phage display reported the generation of five TEM8-specific human antibodies (L1, L2, L3, L5 and 1D2) which inhibited tumour-induced angiogenesis as well as cancer growth in several mouse tumour models[9]. More recently, the safety and therapeutic potential of a TEM8-specific CAR based on the L2 antibody was explored using mouse models of triple negative breast cancer[21]. These studies showed that not only could infusion of these L2 CAR T-cells inhibit tumour growth, but there were no toxic effects reported.

Here we present contrasting findings using a panel of TEM8-specific CARs based on the same antibodies. Some of these, including L2, caused significant toxicity in healthy mice, apparently through targeting of TEM8 in healthy tissue, thus raising concerns over the use of such CARs for human studies.

## Results

### Generation and expression of TEM8-specific CARs

The gene sequences encoding scFv domains based on the TEM8-specific human antibodies L1, L2, L3, L5 and 1D2 were cloned into retroviral vectors to encode second generation CARs

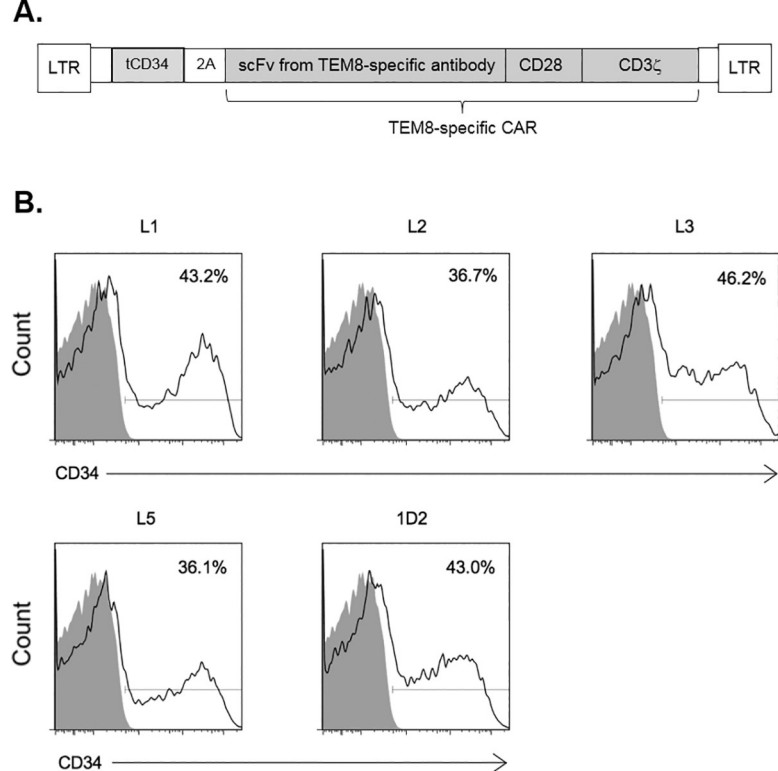

**Fig 1. Generation and expression of CAR constructs in T-cells.** (A) Schematic representation of recombinant retroviral vectors encoding TEM8-specific CARs. Recombinant retroviral vectors coexpressed a truncated human CD34 marker and an scFv fragment from one of 5 different TEM8-specific monoclonal antibodies linked to a CD28 costimulatory domain and a CD3 zeta chain. The 2A peptide linker ensured equimolar expression of both gene constructs. (B) Expression of all 5 TEM8-specific CARs (based on monoclonal antibodies L1, L2, L3, L5 and 1D2) in T cells was demonstrated by flow cytometry staining for the coexpressed CD34 marker. % values show proportion of cells stained for CD34 in transduced T cells (black line) compared to mock-transduced T cells (shaded).

with a CD28 costimulatory domain upstream of the CD3ζ chain as illustrated in Fig 1A. In addition, the CAR gene was separated from a truncated CD34 marker gene by a foot-and-mouth disease virus 2A peptide linker, ensuring equimolar expression of both genes. In this way CD34 expression could be used as a marker for CAR expression. Following transduction of human T-cells using these retroviral vectors, the CD34 marker gene was clearly detectable by flow cytometry for all five CAR constructs, indicating that all five CARs are readily expressed (Fig 1B).

## In vitro testing of TEM8-specific CAR T-cells

To explore the function of human T-cells expressing the five different CARs, they were co-cultured with the LS174T tumour cell line which is naturally TEM8-negative[9] but which we engineered to stably express the mouse form of TEM8 (S1 Fig). As shown in Fig 2A, when testing for secretion of interferon-gamma (IFN-gamma), T-cells expressing four out of the five CARs specifically responded to the TEM8-expressing LS174T cells, with the strongest responses seen with L1 and L2 CARs. The only CAR that failed to show a clear response was that based on the 1D2 antibody. Further studies explored the cytotoxic activity of the different CAR T-cells using the same target cells and showed a similar pattern of response, with the highest levels of cytotoxicity seen with L1, L2 and L3 CARs, but no specific cytotoxicity with

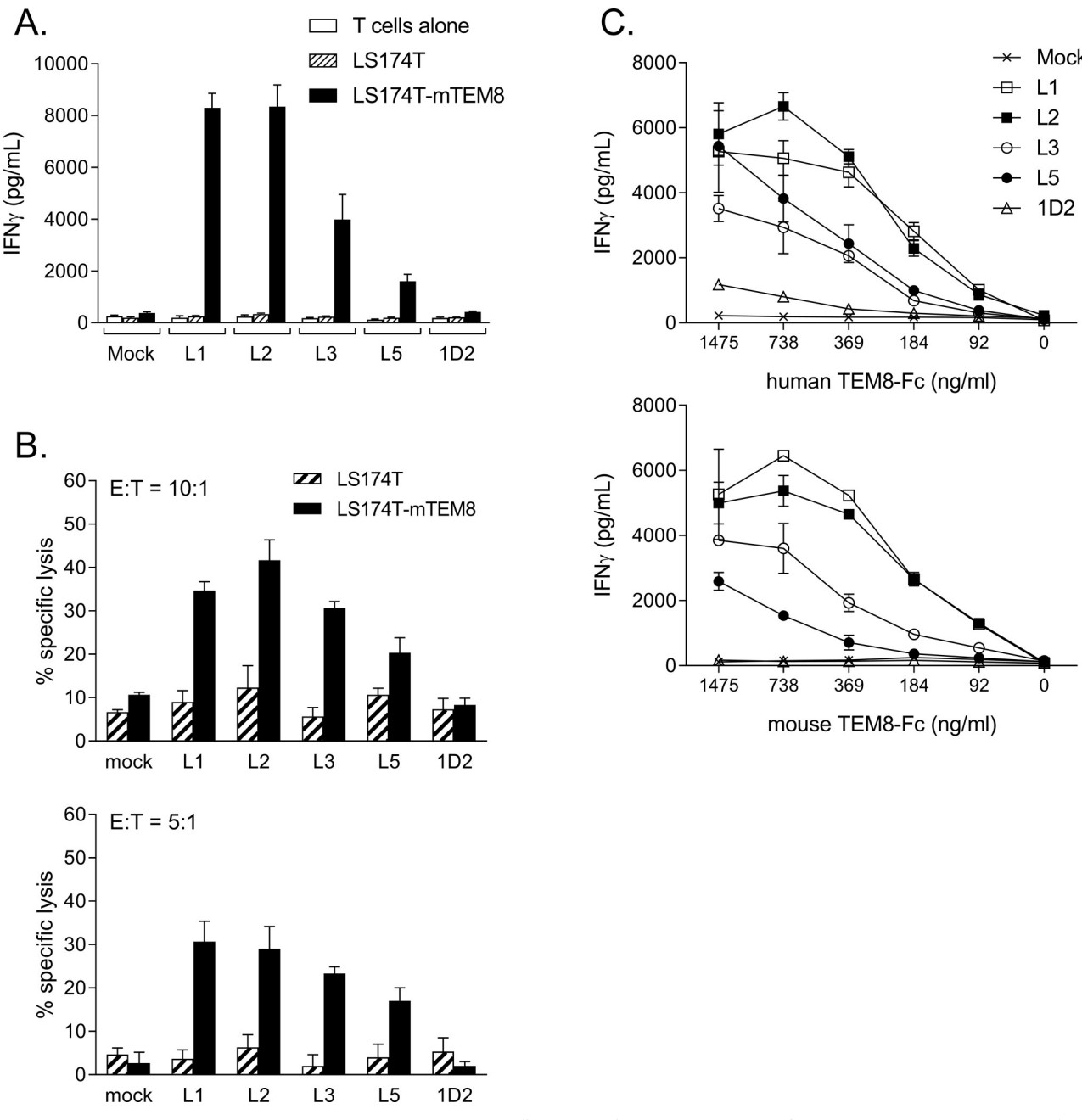

**Fig 2. CAR T cell responses to mouse and human TEM8.** Human T cells engineered to express TEM8-specific CARs (L1, L2, L3, L5, 1D2) or mock-transduced T cell controls were tested for response to LS174T cells engineered to express full-length mouse TEM8. Antigen specific responses were detected using (A) IFN-gamma ELISA and (B) cytotoxicity assay (tested at effector:target (E:T) ratios of 10:1 and 5:1). (C) Responses to titrated concentrations of recombinant human and mouse TEM8 protein were assessed by IFN gamma release. In all cases CAR-T cell lines were diluted with mock T cells to equalise for transduction efficiency. Graphs show the mean of triplicate cultures (± standard deviation, SD) and are each representative of 3 experiments.

the 1D2 CAR (Fig 2B). Using recombinant human and mouse TEM8 proteins, titration studies showed that the CAR T-cell response (measured by secretion of IFNγ) was strongest with L1 and L2 CARs, and somewhat weaker responses seen with L3 and L5 CARs. However, although the 1D2 CAR showed low but detectable responses to the human TEM8 protein when tested at

higher concentrations, there was no detectable response to the mouse protein (Fig 2C). These results compare well with the binding affinities of these antibodies, since the affinities of L1, L2, L3 and L5 for human and mouse TEM8 are all within the range of 0.18–0.36nM. In contrast, the affinity of 1D2 for human TEM8 is only 7.6nM, and for mouse TEM8 it is 31.2nM, approximately 100 times lower than each of the other four antibodies[9]. Data demonstrating the expression and antigen-specific function of these CARs in mouse T-cells are shown in S2 Fig.

## In vivo testing of TEM8-specific CAR T-cells

Having demonstrated TEM8-specific function for four of the CARs, we then began a series of small toxicity studies in healthy mice. In the first experiment, C57BL/6 mice were treated with a single intravenous infusion of 20 million total T-cells derived from a congenic (CD45.1$^+$) mouse strain with 55%, 64%, 40% or 55% of these cells expressing the L1, L2, L3 or L5 CARs respectively. Not only were mice monitored closely for signs of toxicity, but the persistence of the infused CAR T-cells was also monitored by serial tail bleeds using the CD34 marker as an indicator of CAR expression and CD45.1 as a marker of the infused T-cell population. After 24hrs, mice injected with the L2 and L3 CAR showed signs of toxicity with a hunched posture, piloerection and greatly reduced levels of activity which necessitated culling of the animals. The remaining mice treated with L1 and L5 CARs did not show such signs, but tail bleeds indicated that although more than half of the infused T-cells initially expressed the CAR, within 3 days post-infusion nearly all of the L1 and L5 CAR-expressing cells were no longer detectable in the circulation. In contrast, infused T-cells that did not express the CAR (i.e. CD34$^-$ CD45.1$^+$) were clearly detectable and represented 35–42% of the total circulating T-cell pool at 3 and 7 days post infusion. The selective loss of CAR-expressing T-cells continued until day 28 when the experiment was ended (Fig 3A).

In the second in vivo experiment, congenic mouse T-cells expressing the L1 or L5 CARs were again tested in healthy C57BL/6 mice but this time we also tested the 1D2 CAR, and T-cells expressing a control CAR that lacked a scFv domain (no scFv) and therefore should not target an antigen. Mice were infused with 20 million total T-cells of which 81%, 70%, 84% and 77% expressed the L1, L5, 1D2 and no scFv CARs respectively. None of the mice showed signs of overt toxicity, but all mice treated with the L1 and L5 CARs again showed rapid and selective depletion of the CAR expressing T-cells such that these cells became almost undetectable from day 9 (Fig 3B). In contrast, T-cells infused into these L1 and L5-treated mice that did not express the CAR (i.e. CD34$^-$ CD45.1$^+$) were again clearly detectable and represented >35% of the total circulating T-cell pool. Mice treated with T-cells expressing the no scFv control CAR showed an initial drop in the percentage of infused cells expressing this CAR, but this stabilised 4 days post-infusion and remained at ~50% for the next 17 days. 1D2 CAR T-cells were selectively depleted during the first week but then remained at ~20% for the rest of the experiment.

The third in vivo experiment used human T-cells injected into healthy immunocompromised NSG mice. Three mice received 20 million total T-cells of which 21% expressed the L2 CAR and again within 24hrs they all showed the same signs of toxicity seen in the first in vivo experiment, which necessitated culling of the animals. In contrast, mice (n = 3 per group) treated with 20 million total human T-cells expressing the 1D2 CAR or the no scFv CAR control showed no signs of toxicity even when infused again on day 5 and day 9 with the same cells. Note 1D2 and no scFv CARs were expressed on 25% and 22% of the infused T cells respectively. Tail bleed analysis again showed that the 1D2 CAR T-cells persisted in the circulation to the same extent as the no scFv control CAR (Fig 3C). The fourth in vivo experiment largely repeated the third although L2 and no scFv CARs were expressed on 56% and 63% of

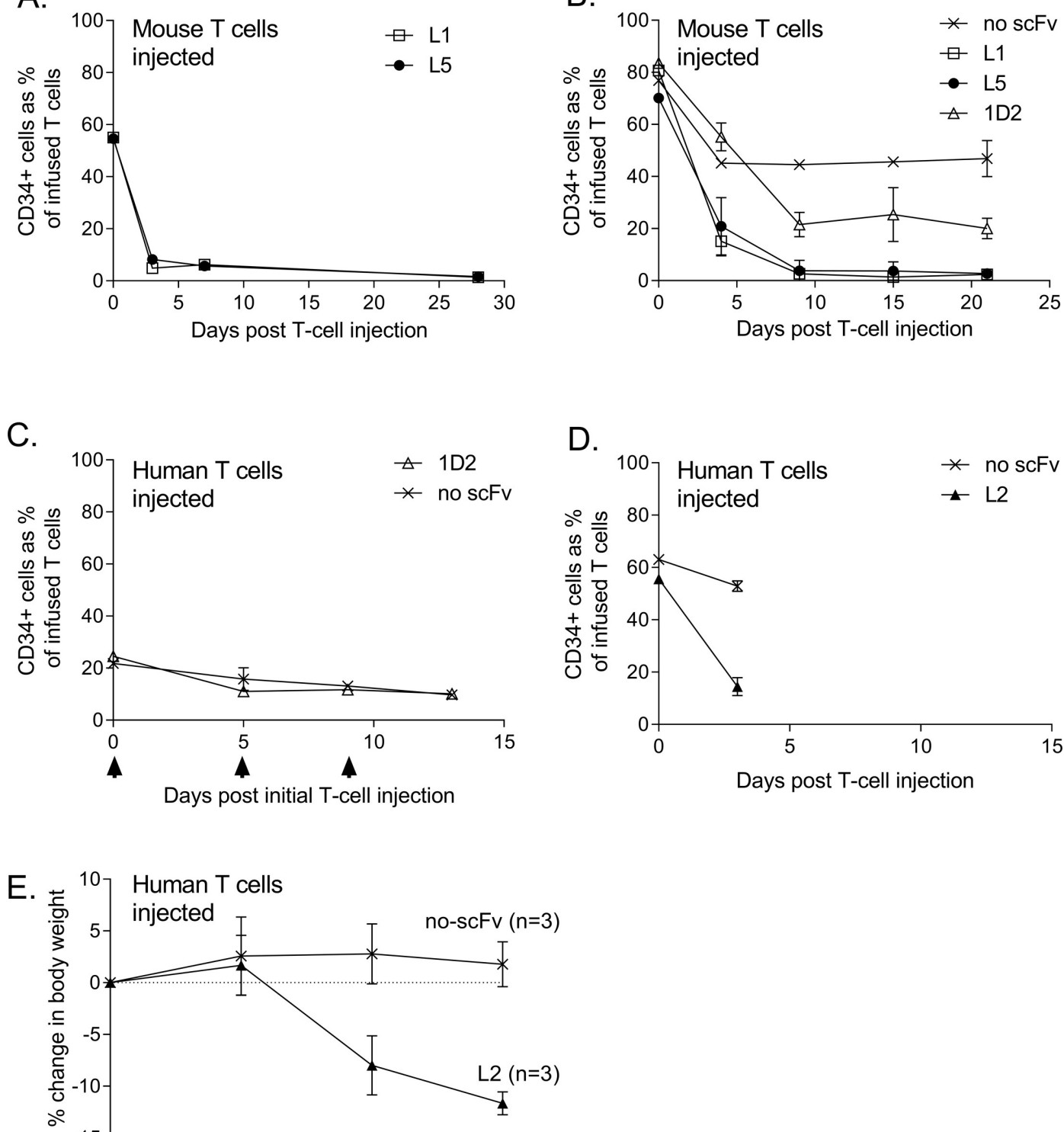

**Fig 3. TEM8-specific CAR T cells are selectively lost from the circulation in healthy mice.** (A) Mouse T cells engineered to express TEM8-specific CARs (L1, 55% transduced; L2, 64% transduced; L3, 40% transduced; L5, 55% transduced) or mock-transduced T cell controls were injected (20 million total T cells/mouse) into single healthy C57BL/6 mice and persistence measured by sequential tail bleeds. Note mice treated with L2 and L3 CARs are not shown as they had to be culled

within 24hrs post treatment. (B) Mouse T cells engineered to express TEM8-specific CARs (L1, 81% transduced; L5, 70% transduced; 1D2, 84% transduced) or a control CAR (no scFv, 77% transduced) were injected (20 million total T cells/mouse) into healthy C57BL/6 mice (n = 2 per group) and persistence measured by sequential tail bleeds. (C) Human T cells engineered to express TEM8-specific CARs (L2, 21% transduced; 1D2, 25% transduced) or a control CAR (no scFv, 22% transduced) were injected (20 million total T cells/mouse) into healthy NSG mice (n = 3 per group) on 3 occasions (indicated with black arrowheads) and persistence measured by sequential tail bleeds. Note mice treated with L2 CARs are not shown as they had to be culled 24hrs post treatment. (D) Human T cells engineered to express TEM8-specific CARs (L2, 55.6% transduced) or a control CAR (no scFv, 63% transduced) were injected (20 million total T cells/mouse) into healthy NSG mice (n = 3 per group) and persistence measured by tail bleeds. Changes in body weight of these mice are shown in (E). Graphs A-D show CAR T cells (CD34+) as a percentage of the total infused T cell population. Results shown in B are the mean of duplicate values and results shown in C, D and E are the mean of triplicate values (± SD).

20 million total infused T-cells respectively, and 1D2 CAR was not tested. Toxicity was again observed in all three L2 CAR-treated mice (hunched posture, piloerection and greatly reduced levels of activity) this time 3 days post infusion. In contrast, mice treated with the control (no scFv) CAR showed no toxicity. Three days post infusion, mice were analysed for persistence of infused cells and again there was clear evidence for selective loss of TEM8-specific CAR T-cells from the peripheral blood (Fig 3D). The L2 CAR-treated mice in this fourth in vivo experiment also showed a clear loss in body weight whereas no such change was seen in control (no scFv) CAR treated mice (Fig 3E). Three days post infusion all mice in this fourth experiment were culled and tissues taken for histological analysis. Haematoxylin and eosin stained sections were analysed by an experienced pathologist who was unaware of the treatment received by each mouse. Brain, colon, liver, kidney, pancreas, and heart tissues showed no pathology in either group of mice, but lung and spleen tissues showed evidence of inflammation in all L2 CAR treated animals, with numerous neutrophils present both in the red pulp of the spleen and surrounding the blood vessels and bronchioles in the lung. In contrast, there was no infiltration of neutrophils in the lung or spleen tissues of any mice treated with the control (no scFv) CAR (Fig 4 and S3 Fig).

Finally, to explore whether these effects were dependent on TEM8 expression, we used a TEM8 knockout mouse. Mouse T-cells from a congenic strain were engineered to express the L2 TEM8-specific CAR or a control (no scFv) CAR and injected intravenously into healthy wild type and TEM8 knockout C57BL/6 mice. On this occasion, the L2 CAR T-cells did not mediate a toxic reaction even though the number of infused T-cells that expressed the CAR was similar to previous experiments. Nevertheless, these CAR T cells were again selectively lost from the circulation in wild type mice as had been observed with all the high affinity TEM8--specific CARs in the mouse experiments described above. Importantly, however, L2 CAR T-cells persisted in the circulation of TEM8 knockout mice. The control CAR T-cells persisted in the circulation of both mouse strains. Thus selective loss of L2 CARs from the circulation was TEM8-dependent (Fig 5).

## Discussion

Studying a panel of CARs specific for the tumour endothelial marker TEM8, we identified significant toxicity when some of them were tested in healthy mice. Furthermore, we also demonstrated selective and rapid depletion of all high avidity TEM8-specific CAR T-cells from the circulation. This suggested the CARs were targeting TEM8 or cross-reacting with another antigen expressed on healthy tissue(s) which resulted in toxicity and/or migration of the CAR T-cells out of the bloodstream and into the target tissue(s). Histological analysis of TEM8-specific CAR T-cell-treated mice also revealed evidence of inflammation in lung and spleen tissue. Studies using a TEM8 knockout mouse subsequently showed that selective loss of the CAR T-cells from the circulation was TEM8 dependent, strongly suggesting that the CAR T-cells are recognising TEM8 expressed on healthy tissue(s). Thus, despite elevated levels of TEM8

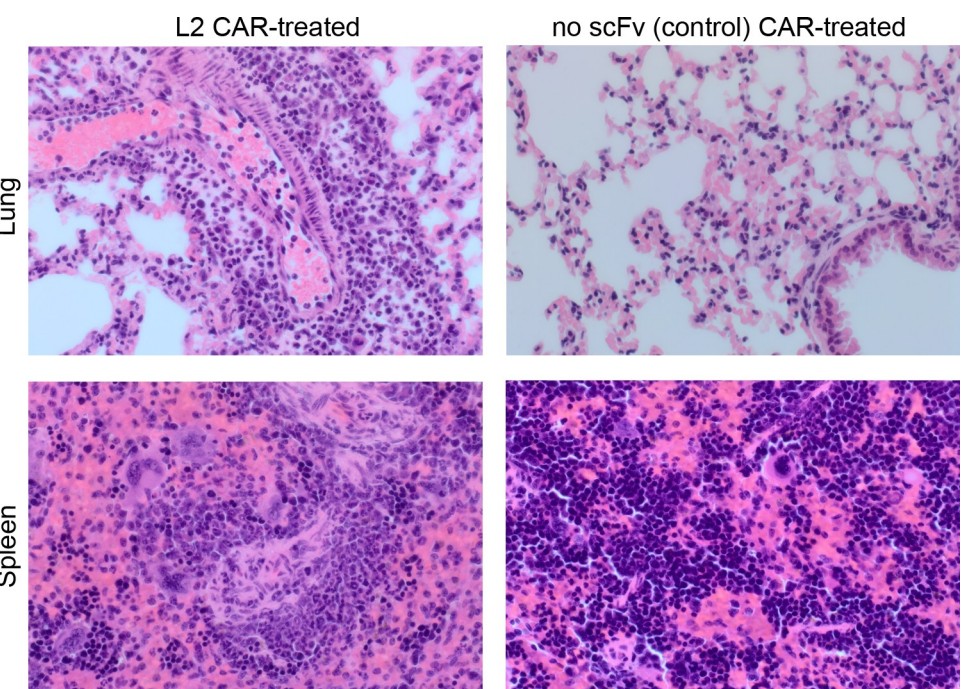

**Fig 4. Histological analysis of mice treated with TEM8-specific or control CAR T-cells.** Representative images of haematoxylin and eosin stained lung and spleen tissues from the fourth in vivo experiment where NSG mice were treated with human T cells engineered to express TEM8-specific CARs (L2) or a control CAR (no scFv). Mice (n = 3 per group) were injected with 20 million total T cells/mouse of which 11.1 million and 12.6 million expressed the L2 and no scFv CARs respectively. Tissues were taken 3 days later and showed evidence of neutrophil infiltration in all L2 CAR treated mice but none of the control CAR treated mice. (Magnification = x400).

expression in tumour tissues, levels on healthy tissues can still be sufficient to be targeted by a potent agent such as a high avidity multivalent CAR T-cell.

This finding is particularly significant given a recent publication demonstrating safety and anti-tumour efficacy in a mouse model of triple negative breast cancer using a CAR based on the same L2 antibody that mediated toxicity in 7/11 wild type mice in our studies[21]. Reasons for these contradictory findings could lie in subtle changes in the design of the CAR. For example, different retroviral vectors may have led to different levels of CAR expression per cell. Also, the scFv domain in our study comprised the heavy chain antibody sequence followed by the light chain, whereas in the other study the genes encoding these antibody chains were in reverse order. Equally, the other study tested two versions of L2 CAR that included a costimulatory region from CD28 with or without an additional costimulatory domain from 4-1BB. Most of their in vivo studies focused on the second CAR and showed no toxicity using doses of up to 10 million cells (70% transduced). Nevertheless, no toxicity was reported when the CAR with the CD28 domain alone was injected intratumourally at a dose of 25 million cells/mouse. Whatever the reasons for the lack of toxicity in this other study, our results demonstrating toxicity with an L2 CAR even at doses of 4 million CAR-expressing T-cells per mouse, combined with evidence for TEM8-mediated selective loss of high avidity CAR T-cells from the circulation of healthy mice, highlights the risk of on-target, off-tumour effects should a TEM8-specific CAR, especially one based on the L2 antibody, be tested in patients. This is of particular concern if strategies are employed to enhance the anti-tumour effects of the CAR, for example through dose escalation or increasing the levels of CAR expression and/or function. In support of our findings, it should be noted that toxicity was also reported when the L2 antibody was

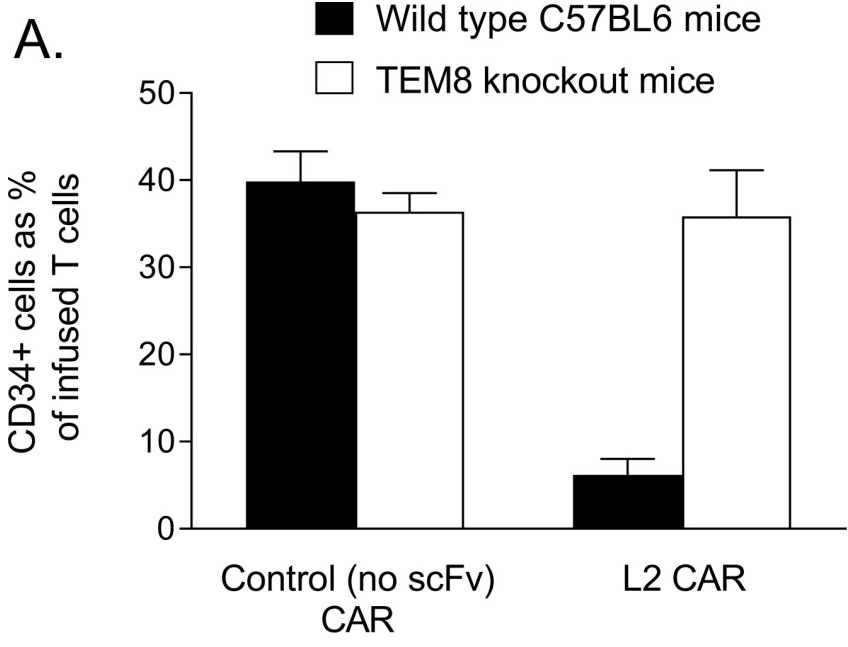

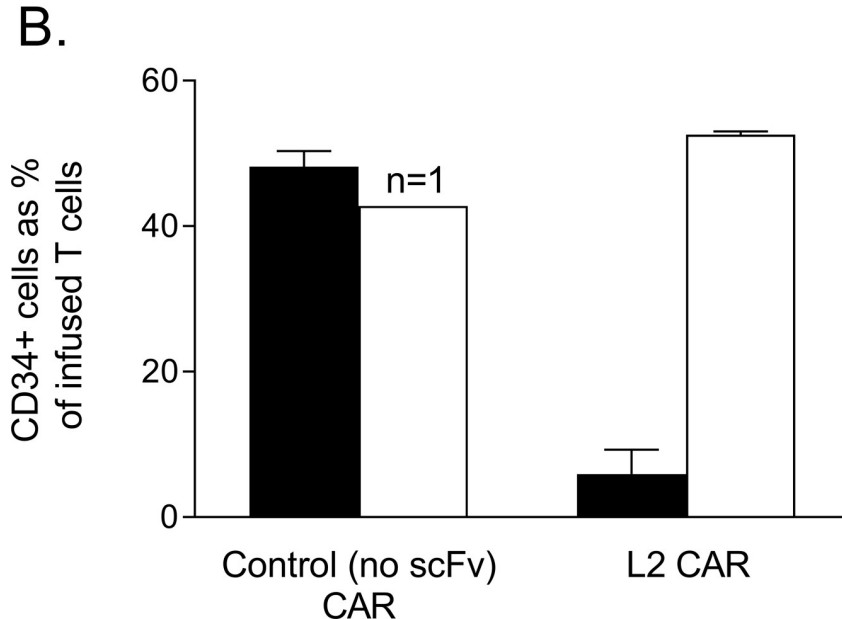

**Fig 5. Persistence of TEM8-specific CAR T cells in the circulation of healthy wild type and TEM8 knockout mice.**
Mouse T cells from a congenic strain were engineered to express the L2 TEM8-specific CAR or a control CAR (no scFv) lacking the scFv antigen-binding domain. T cells were injected intravenously into wild type (black bars) or TEM8 knockout mice (white bars) and tail bleeds sampled to identify the infused cells (CD45.1[+]) that expressed the CAR (CD34[+]). Graphs show the results of two separate experiments where tail bleeds were analysed (A) 8 days and (B) 6 days post infusion. All experiments used two mice per group (except where indicated). T cells engineered for these experiments were >50% transduced, and for data shown in (A) and (B) the number of infused T cells per mouse that actually expressed a CAR was 5 million and 12 million respectively. Data represent the mean percentage (+ SD) of infused cells that expressed a CAR.

used to engineer mouse T-cells to express a secretable bi-specific T-cell engager. This engager used a TEM8-specific scFv from the L2 antibody linked to a scFv recognizing CD3. Intravenous injection of 10 million of these T-cells resulted in death of 7/10 mice[22].

To reduce the risk of on-target, off-tumour toxicity with such CARs, one possibility is to use them in a combinatorial approach, where T-cells are engineered to express two distinct receptors, and activation is dependent on encountering both target antigens which are only co-expressed in tumour tissues[23, 24]. Use of RNA transfection of T-cells for transient expression of the CAR[25] or use of suicide gene strategies[26] can also limit the risks, or better still a "tunable" system where CAR expression levels can be regulated through administration of a drug[27]. Alternatively, reducing the affinity of the TEM8-specific CAR might permit discrimination between levels of the target antigen expressed at the tumour site and levels expressed in healthy tissues as has been shown for other CARs[28, 29]. In this regard it is interesting to note that although the 1D2 TEM8-specific antibody has a 100-fold lower affinity than L2 for mouse TEM8[9], and according to our data T-cells expressing a 1D2 CAR do not respond to the mouse protein, this antibody has intermediate affinity for human TEM8. Preclinical safety and efficacy testing of a 1D2 CAR in a mouse model would, however, require generation of a human TEM8 knockin.

## Methods

### Generating human and mouse CAR T cells

The gene sequences encoding each of the TEM8-specific human antibodies L1, L2, L3, L5 and 1D2 were kindly provided by B. St Croix (National Cancer Institute, Frederick, MD) and used to design synthetic DNA sequences (GenScript) encoding an scFv region and then cloned into the CAR vector pMP71.tCD34.2A.CD19.IEVζ[30] as a ClaI, NotI fragment, replacing the CD19-specific scFv region. These vectors were originally constructed using the MP71 retroviral expression plasmid (a kind gift from C. Baum, Hannover) and co-expressed a truncated human CD34 marker gene[31]. The gene sequences of all new constructs were verified.

To generate recombinant retrovirus for transducing mouse T cells, Phoenix ecotropic packaging cells were transfected with the appropriate MP71 retroviral expression vector encoding the CAR and pCL eco (Imgenex) using FuGENE HD (Roche) according to the manufacturer's instructions. Recombinant retrovirus for transducing human T cells was generated in the same way but using Phoenix amphotropic packaging cells and pCL ampho. Transduction of mouse T cells was conducted using mouse splenocytes pre-activated for 48 hours with concanavalin A (2μg/ml; Sigma) and mouse interleukin-7 (1ng/ml; eBioscience) in RPMI 1640 (Sigma) containing 10% foetal bovine serum (FBS; PAA, Pasching Austria), 2mM L-glutamine, 100 IU/ml penicillin, and 100ug/ml streptomycin (standard medium). For human T cells, peripheral blood mononuclear cells (PBMCs) were isolated from aphereses cones[32] collected from healthy adult volunteers (male or female, aged >18 years) who attended the National Blood Donor Centre, Birmingham. PBMCs were isolated by density gradient centrifugation on Lymphoprep (Axis Shield, Oslo, Norway). They were then pre-activated for 48 hours using anti-CD3 antibody (30ng/ml; OKT3 eBioscience), anti-CD28 antibody (30ng/ml; 37407 R&D Systems) and interleukin-2 (IL2, 300 U/ml; Chiron, Emeryville, CA) in standard medium containing 1% human AB serum (TCS Biosciences, Buckingham, UK). Pre-activated human and mouse T cells were subsequently transduced (or mock-transduced with conditioned supernatant from non-transfected Phoenix cells) by spinfection in retronectin (Takara)-coated plates according to the manufacturer's instructions. After spinfection, mouse T cells were cultured for 24hrs in standard medium with IL2 (100 U/ml), then purified using Ficoll-Paque premium 1.084 (GE Healthcare). Human T cells were cultured in standard medium plus 1% human AB serum with IL2 (100 U/ml).

## Cell lines

LS174T cells[33] were obtained from ATCC and cultured in DMEM containing 10% FBS, 2mM L-glutamine, 100 IU/ml penicillin, and 100μg/mL streptomycin. Stable expression of mouse TEM8 within these cells was achieved by transduction using the lentiviral plasmids psPAX2 (lentiviral packaging; Addgene, Cambridge, MA, USA), pMD2G (envelope plasmid; Addgene) and pWPI lentiviral mammalian expression plasmid (Addgene) into which we had cloned the mouse TEM8 gene following digestion using PacI and PmeI restriction enzymes (New England BioLabs, MA). Plasmids were transfected into 293T cells (from ATCC) using FuGENE 6 according to the manufacturer's instructions in a ratio of 1:2.5:3.3 for pMD2G: psPAX2:pWPI. LS174T cells were then transduced with lentivirus-containing culture supernatant mixed with polybrene (8μg/ml, Merck, Millipore). Cell lines were screened for mycoplasma using MycoAlert detection kit (Lonza, Basel).

## Recombinant TEM8

Genes encoding the human or mouse forms of TEM8 fused to the human Fc protein were cloned into the pFUSE-mIgG2A-Fc1 vector (InvivoGen, San Diego, CA). TEM8 protein production was then carried out with the help of the Protein Expression Facility, University of Birmingham. Human TEM8-Fc/pFUSE-mIgG2A or mouse TEM8-Fc/pFUSE-mIgG2A plasmids were transfected into 293T cells using polyethylenimine (PEI 25000; Polysciences, Warrington, PA) in Opti-MEM Reduced Serum Medium (Gibco) and 7 days later supernatants were harvested and protein purified using Protein A Sepharose CL-4B (GE Healthcare) according to the manufacturer's instructions.

## IFNγ ELISA

LS174T cells ($2x10^4$/well) were co-cultured in triplicate with human CAR T-cells ($2x10^5$/well) in 96-well flat bottom plates. Alternatively CAR T-cells ($2x10^5$/well) were incubated in wells pre-coated for 4 hours with recombinant TEM8-Fc protein (or Fc protein alone) at concentrations indicated. Cells were incubated at 37°C/5% $CO_2$ in 100μl/well of RPMI supplemented with 10% FBS, 2mM L-glutamine, 100 IU/ml penicillin, and 100ug/ml streptomycin and IL2 (25 U/ml). After 18 hours, culture supernatant was tested for secreted IFN-gamma using an ELISA (Pierce Endogen, Rockford, IL) according to the manufacturer's instructions.

## Cytotoxicity assay

Chromium release assays have been described in detail previously[34]. They were set up at known effector:target ratios (1250 targets/well) and harvested after 7.5 hours.

## In vivo studies

Mice were housed in individually ventilated containers with a 12 hour day/night light cycle at temperatures of 21 +/- 2°C and relative humidity of 55% +/- 5%. All mice were allowed free access to water and a maintenance diet (EURodent diet 14%). All cages contained wood shavings, bedding material and a plastic house. Six to eight week old female C57BL/6 mice (Charles River Laboratories) or TEM8-knockout mice[17] (a kind gift from B. St Croix, National Cancer Institute, Frederick, MD) received non-myeloablative (5 Gy) total body irradiation (TBI) and 18 hours later CAR- or mock-transduced T-cells from CD45.1[+] congenic BoyJ mice (Charles River Laboratories) were injected into the tail vein. Six to eight week old female NSG mice (Charles River Laboratories) were injected into the tail vein on up to 3 occasions with 20 million CAR- or mock-transduced human T-cells at the times and doses indicated. All mice

were monitored every 15 minutes for the first hour post infusion, and then at least once daily for signs of toxicity. Humane endpoints included >20% body weight loss, hunched posture, continued piloerection and/or inactivity for a period of 24hrs or dyspnoea. Once humane endpoints were reached they were culled immediately (note no animals died before meeting the humane endpoint criteria). Mice were assessed for toxicity by experienced animal workers and to avoid subjective bias they were blinded to the treatment group to which the animal had been assigned. Immune monitoring was conducted with serial tail bleeds. When staining T cells from heparinized tail bleeds they were first subjected to red blood cell lysis using BD Pharm Lyse (Becton Dickinson). They were then washed with PBS and stained with Live/Dead Fixable Violet Dead Cell Stain Kit (Life Technologies) for 20 mins in the dark, then washed with flow buffer (0.5% w/v BSA + 2mM EDTA in PBS; pH7.2). If staining for infused mouse T-cells, they were then stained with anti-mouse CD4-PE (clone H129.19), CD8-FITC (clone 53–6.7) and CD45.1-PE-Cy7 (clone A20) (all from BD Biosciences) for 30mins on ice in the dark. Alternatively when analysing infused human T-cells, they were stained with anti-human CD4-FITC (clone RPA-T4, BD Biosciences) and anti-human CD8-PE (clone RPA-T8, eBioscience). In all cases CAR-expressing cells were identified by co-staining with anti-human CD34-APC (clone 561, BioLegend). Cells were analysed using a BD LSR II flow cytometer and FlowJo software (TreeStar Inc, Ashland, OR). Histological analysis was conducted on formalin-fixed tissue stained with haematoxylin and eosin. Tissues were analysed using an Olympus SC100 microscope with Olympus cellSens Standard software.

## Study approval

Studies with human donors were approved by the National Research Ethics Service Committee West Midlands (Solihull, UK). All donors were adults and gave written informed consent. All mouse studies were performed under UK Home Office authorisation and all animal workers were fully trained and held personal licenses granted by the UK Home Office.

## Supporting information

**S1 Fig. Mouse TEM8 expression in engineered LS174T cells.** LS174T cells transduced with a lentivirus expressing the murine form of TEM8 were stained with the TEM8 specific L2 monoclonal antibody (80ug/ml) or a concentration- and isotype-matched antibody control (clone ZX4, Thermo Fisher Scientific). They were then stained with a phycoerythrin-conjugated goat anti-mouse IgG antibody (Biorad) and analysed by flow cytometry using an LSRII Cytometer (Becton Dickinson) and FlowJo software (Tree Star).
(TIF)

**S2 Fig. CAR expression and function in engineered mouse T-cells.** (A) Expression of all 5 TEM8-specific CARs (L1, L2, L3, L5 and 1D2) and the no scFv control CAR in mouse T cells was demonstrated by flow cytometry staining for the coexpressed CD34 marker. % values show proportion of cells stained for CD34 in transduced T cells (black line) compared to mock-transduced T cells (shaded). (B) Antigen specific responses to LS174T cells expressing mouse TEM8 were detected using a mouse IFN-gamma ELISA platinum kit (Invitrogen). CAR-T cell lines were diluted with mock-transduced T cells to equalise for transduction efficiency. The graph shows the mean of duplicate cultures (± standard deviation, SD).
(TIF)

**S3 Fig. Representative images of haematoxylin and eosin stained tissues from the fourth in vivo experiment where NSG mice were treated with human T cells engineered to express TEM8-specific CARs (L2) or a control CAR (no scFv).** Mice (n = 3 per group) were injected

with an effective dose of 11.1 million or 12.6 million T cells that all expressed the L2 or no scFv CAR respectively. Tissues were taken 3 days later. (Magnification = x200).
(TIF)

## Author Contributions

**Conceptualization:** Steven P. Lee.

**Data curation:** Steven P. Lee.

**Formal analysis:** Kristina Petrovic, Abeer Shaaban, Steven P. Lee.

**Funding acquisition:** Kristina Petrovic.

**Investigation:** Kristina Petrovic, Joseph Robinson, Katharine Whitworth, Elizabeth Jinks, Abeer Shaaban, Steven P. Lee.

**Project administration:** Steven P. Lee.

**Supervision:** Joseph Robinson, Steven P. Lee.

**Writing – original draft:** Kristina Petrovic, Steven P. Lee.

**Writing – review & editing:** Kristina Petrovic, Joseph Robinson, Katharine Whitworth, Elizabeth Jinks, Abeer Shaaban, Steven P. Lee.

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
