## [Decision Letter · Decision Letter 0]

29 Aug 2019

[EXSCINDED]

PONE-D-19-17502

TEM8/ANTXR1-specific CAR T cells mediate toxicity in vivo

PLOS ONE

Dear Dr. Lee,

Thank you for submitting your manuscript to PLOS ONE. After careful consideration, we feel that it has merit but does not fully meet PLOS ONE’s publication criteria as it currently stands. Therefore, we invite you to submit a revised version of the manuscript that addresses the points raised during the review process.

Please make sure to address all comments by reviewers and make necessary changes.

We would appreciate receiving your revised manuscript by 9/21/19. To enhance the reproducibility of your results, we recommend that if applicable you deposit your laboratory protocols in protocols.io, where a protocol can be assigned its own identifier (DOI) such that it can be cited independently in the future. For instructions see: http://journals.plos.org/plosone/s/submission-guidelines#loc-laboratory-protocols

We look forward to receiving your revised manuscript.

Kind regards,

Nupur Gangopadhyay, B.V.Sc, M.V.Sc.,Ph.D.

Academic Editor

PLOS ONE

**Journal Requirements:**

"I have read the journal's policy and the authors of this manuscript have the following competing interests:  S.Lee is listed as an inventor on the following patent application/patent families; PCT/GB2017/050686 (published as WO2017/158337) and PCT/GB2017/050689 (WO2017/158339), and has interests in anti-CLEC14A CAR-T cells, studies relating to which have been supported by Cell and Gene Therapy Catapult UK and licensed to Chimeric Therapeutics Ltd"

**Comments to the Author**

1. Is the manuscript technically sound, and do the data support the conclusions?

Reviewer #1: Partly

Reviewer #2: Yes

2. Has the statistical analysis been performed appropriately and rigorously? 

Reviewer #1: Yes

Reviewer #2: Yes

3. Have the authors made all data underlying the findings in their manuscript fully available?

Reviewer #1: Yes

Reviewer #2: Yes

4. Is the manuscript presented in an intelligible fashion and written in standard English?

Reviewer #1: Yes

Reviewer #2: Yes

5. Review Comments to the Author

Reviewer #1: Petrovic et al. describe a generally well done set of studies showing significant toxicity with CARs made with a series of anti-TEM8 antibodies. These data are in contrast to a recently published a paper in Cancer Research by the Baylor group that showed efficacy but no toxicity.

CAR behaviors can be very construct dependent, with slight changes making big differences. It is not entirely clear why the results of these two studies are different, but this is discussed in the discussion. However, I am very much in favor of publishing studies that show potential toxicity of CARs, as this needs to be taken into account when designing future clinical trials. The Baylor CAR may not be toxic, but certainly this paper provides data to suggest careful toxicity studies and a cautious clinical trial design are needed.

The validity of their findings is strengthened in my mind by NOT seeing toxicity in a Tem8 knockout mouse.

There are some critical omissions/clarifications that need to be addressed however.

1. While transduction efficiency and killing assays are shown for the human CAR T cells in figs 1 and 2, no data is shown for the mouse CAR T cells. This is critical to provide, as mouse CAR T cells are difficult to make, and form a large part of the in vivo data.

2. It would also be important to show the expression levels of Tem8 on the LS174T tumor cells. Were these cells sorted?

3. Related to this- the authors must clarify an important point: In their animal experiments they state that they injected “20 million congenic mouse T cells expressing the CAR”. Does this mean they inject 20 million total T cells or that they injected 20 million CAR expressing T cells? Here is where the transduction efficiency is so important. If they injected 20 million CAR expressing T cells, depending on the transduction efficiency, this could be a very large number of cells (i.e. a 50% transduction efficiency would mean injection of 40 million T cells). We usually do not inject greater than ~20 million total T cells.

4. There was no histology data provided to give any clue about why the mice got so sick after injections. It can be hard to tell why a mouse died, but the paper would be strengthened by this data. Was there inflammation in the lungs, heart, liver, kidney, etc? Was there widespread vasculitis?

Reviewer #2: Summary:

This paper presents contrasting data in terms of toxicity with TEM8-specific CARs to that which has been previously published. They show development of 5 different TEM8 CAR T cell variants generated from human L1, L2, L3, L5, and ID2 antibodies and their expression and specific cytotoxicity efficacy in vitro via INFγ secretion and chromium release cytotoxicity studies with appropriate controls. ID2 was the only CAR T cell that did not have specific cytotoxicity. They then performed in vivo experiments for persistence by injecting these CAR T cells into healthy immunocompetent mice and performing serial blood draws. The mice injected with L2 and L3 TEM8 CAR T cells had severe toxicity. While the L1 and L5 variants did not have toxicity the cells did not persist long. Finally they demonstrate that the failure to persist is most likely related to recognition of TEM8 on healthy tissue as persistence is of these cells is demonstrated in the TEM8 knockout mice.

Comments:

• The n for mouse studies is small with only 2-3 mice per group, however, results are consistent between the mice in each study

• The timing of the blood draws for CAR T cell persistence is very variable between experiments and it is not clear why this is the case. The initial experiment was done with blood draws on day 3, 7, and 28. The second experiment looks like day 4, 9, 15, and 21. Then day 3 for the human T-cell experiment which was when these mice needed to be culled. Finally in the knockout mice blood draws happened day 8 for one experiment and day 6 for the other experiment. It is just very inconsistent

• In addition, in the last mouse experiment with the TEM8 knockout mice, the authors comment on the fact that the L2 scFv did not mediate a toxic reaction in this experiment but they also used only 5 million CAR T-cells compared to 20 million all prior experiments with similar transduction rates. I think this should be stated more clearly in the text and not just in the figure legend. The should also justify why they changed the dosing

• The figures could also have some tighter formatting in terms of size balance and labels. Specifically, for Figure 3 it would be helpful to label in the figure which graphs are done with mouse T-cells and which with human T-cells. In Figure 4, a legend showing what the bars represent rather than just stating it in the figure legend text

• Additionally, there are some minor grammatical errors and areas were the wording could be tightened up a bit – specifically in the intro.

6. PLOS authors have the option to publish the peer review history of their article (what does this mean?). If published, this will include your full peer review and any attached files.

Reviewer #1: No

Reviewer #2: No

---

## [Author Response · Author response to Decision Letter 0]

20 Sep 2019

Dear Dr Gangopadhyay

Re. Manuscript PONE-D-19-17502 (“TEM8/ANTXR1-specific CAR T cells mediate toxicity in vivo”).

Thank you for allowing us to submit a revised version of our manuscript and our thanks also to the reviewers for their comments. Each of the points raised by the reviewers is addressed below.

Reviewer #1

1. While transduction efficiency and killing assays are shown for the human CAR T cells in figs 1 and 2, no data is shown for the mouse CAR T cells. This is critical to provide, as mouse CAR T cells are difficult to make, and form a large part of the in vivo data.

We have included in a supplementary figure (fig S2) data demonstrating efficient transduction of mouse T-cells with our TEM8-specific CARs. This figure also includes data demonstrating that these cells are functional since they released IFNg in response to target cells expressing TEM8 (the same type of assay that we used in figure 3 when testing the function of human T-cells expressing these CARs).

2. It would also be important to show the expression levels of Tem8 on the LS174T tumor cells. Were these cells sorted?

In another supplementary figure (fig S1) we have included flow cytometric data demonstrating expression levels of TEM8 in the engineered LS174T cells. The cells were not sorted.

3. Related to this- the authors must clarify an important point: In their animal experiments they state that they injected “20 million congenic mouse T cells expressing the CAR”. Does this mean they inject 20 million total T cells or that they injected 20 million CAR expressing T cells? Here is where the transduction efficiency is so important. If they injected 20 million CAR expressing T cells, depending on the transduction efficiency, this could be a very large number of cells (i.e. a 50% transduction efficiency would mean injection of 40 million T cells). We usually do not inject greater than ~20 million total T cells.

The total number of T-cells injected into a mouse did not exceed 20 million, and the proportion of these that expressed the CAR is indicated by the transduction efficiency stated. Nevertheless, to avoid any confusion, we have altered the text in the manuscript and figure legends to clarify this point.

4. There was no histology data provided to give any clue about why the mice got so sick after injections. It can be hard to tell why a mouse died, but the paper would be strengthened by this data. Was there inflammation in the lungs, heart, liver, kidney, etc? Was there widespread vasculitis?

We have added a new figure (fig 4) and supplementary figure (fig S3) showing H&E stained sections from major organs taken from mice treated with the L2 CAR T-cells (or a no scFv control CAR). These were tissues taken from mice tested in the fourth in vivo experiment from which we also showed data in fig 3D-E). Interestingly these tissues indicate evidence of inflammation in the lungs and spleen of all 3 CAR treated mice since they contained infiltrating neutrophils. No such infiltration was detectable in the control (no scFv CAR treated mice) or in any other tissues studied.

Reviewer #2

1. The n for mouse studies is small with only 2-3 mice per group, however, results are consistent between the mice in each study

We agree the number of mice in each experiment was small but, as stated by this reviewer, the consistency of our findings between experiments supports the conclusions.

2. The timing of the blood draws for CAR T cell persistence is very variable between experiments and it is not clear why this is the case. The initial experiment was done with blood draws on day 3, 7, and 28. The second experiment looks like day 4, 9, 15, and 21. Then day 3 for the human T-cell experiment which was when these mice needed to be culled. Finally in the knockout mice blood draws happened day 8 for one experiment and day 6 for the other experiment. It is just very inconsistent

Again we agree that the timings varied for the different experiments and this was largely due to logistical constraints on when we were able to take the blood samples. Nevertheless, the results demonstrate a consistent pattern of depletion of TEM8-specific CAR T-cells in wildtype mice from the circulation.

3. In addition, in the last mouse experiment with the TEM8 knockout mice, the authors comment on the fact that the L2 scFv did not mediate a toxic reaction in this experiment but they also used only 5 million CAR T-cells compared to 20 million all prior experiments with similar transduction rates. I think this should be stated more clearly in the text and not just in the figure legend. The should also justify why they changed the dosing

The text in our original manuscript has confused Reviewer #2 since these mice were infused with 5 million or 12 million CAR-expressing T-cells (not total T-cells). The potential for confusion in this area was also mentioned by Reviewer #1 (point 3 above) and so we have modified the text throughout the manuscript to ensure clarity. Therefore the dose of CAR expressing T-cells infused in these experiments was not reduced, but was within the range of doses used in the earlier experiments. Again we have altered the text to make this point clear.

4. The figures could also have some tighter formatting in terms of size balance and labels. Specifically, for Figure 3 it would be helpful to label in the figure which graphs are done with mouse T-cells and which with human T-cells. In Figure 4, a legend showing what the bars represent rather than just stating it in the figure legend text

We have altered these figures as requested.

5. Additionally, there are some minor grammatical errors and areas were the wording could be tightened up a bit – specifically in the intro.

We have attempted to improve the text to remove such errors but without more detail we are unable to deduce which parts of the text the referee is referring to.

We would like to thank the reviewers for their comments which we believe has strengthened the manuscript, and trust the revised version is now acceptable for publication.

Yours sincerely,

Steven P. Lee Ph.D.

Senior Research Fellow

---

## [Editor Report · Decision Letter 1]

4 Oct 2019

TEM8/ANTXR1-specific CAR T cells mediate toxicity in vivo

PONE-D-19-17502R1

Dear Dr. Lee,

We are pleased to inform you that your manuscript has been judged scientifically suitable for publication and will be formally accepted for publication once it complies with all outstanding technical requirements.

With kind regards,

Nupur Gangopadhyay, B.V.Sc, M.V.Sc.,Ph.D.

Academic Editor

PLOS ONE
---

## [Editor Report · Acceptance letter]

10 Oct 2019

PONE-D-19-17502R1 

TEM8/ANTXR1-specific CAR T cells mediate toxicity in vivo 

Dear Dr. Lee:

I am pleased to inform you that your manuscript has been deemed suitable for publication in PLOS ONE. Congratulations! Your manuscript is now with our production department. 

With kind regards,

on behalf of

Dr Nupur Gangopadhyay 

Academic Editor

PLOS ONE